# A More Efficient Conditional Private Preservation Scheme in Vehicular Ad Hoc Networks

**Tao Wang ***  **and Xiaohu Tang**

School of Information Science and Technology, Southwest Jiaotong University, Chengdu 610000, China; xhutang@swjtu.edu.cn

**\*** Correspondence: nirro@163.com; Tel.: +86-138-8093-3379

**Abstract:** It is a challenging issue to provide a secure and conditional anonymous authentication scheme in vehicle ad hoc networks (VANETs) with low storage space and computational cost. In 2008, Lu et al. proposed a conditional privacy preservation scheme called efficiency conditional privacy preservation (ECPP) protocol. The ECPP protocol provides conditional privacy preservation to vehicles in VANETs. That is, on one hand vehicles can achieve anonymous authentication in the network, on the other hand, allow to be traced and revoked if necessary. However, ECPP scheme suffers from high computational cost and large storage. In this scheme, an improved protocol based on the concept of ECPP protocol has been proposed to achieve more efficiency conditional privacy preservation (MECPP) scheme in VANETs. Comparing with ECCP, the computational cost of the proposed scheme has been decreased by about 54% while the communication overhead has been reduced by about 10%. At the same time, a lot of storage space has been saved.

**Keywords:** vehicular ad hoc networks; conditional privacy; revocation; anonymous authentication; authentication; security

---

## 1. Introduction

Many people are seriously injured or killed in road traffic accidents due to carelessness, traffic congestion, traffic violations, inadequate road information, increased population and lack of secure infrastructure. Therefore, reducing traffic congestion and enhancing road safety are the issues that many people are most concerned about. With the development of automotive technology and wireless communication technology, VANETs have aroused widespread interest. In VANET, smart vehicles equipped with the on-board device can be connected to each other and surrounding infrastructures easily [1]. Any vehicle can send other nearby vehicles about the traffic and road conditions to warn of potential emergencies and traffic jams. In addition to helping prevent accidents, VANETs also provide convenience and business services that will help improve a driver's experience [2].

However, before taking this wonderful application into practice, several obstacles in VANETS need to be addressed: delay, service cost, security and privacy. It is worthwhile noting that security and privacy issues are becoming more and more import in many people's lives [3]. So far, the security issues of VANETs have been studied in detail, while the privacy issues still have many open questions [4–6]. In the absence of privacy protection, the adversary can track the location of the target vehicle by collecting their routine information. Even worse, if a legitimate anonymous vehicle becomes malicious, there is no way to identify and revoke it. Thus, it is necessary to limit malicious vehicles, the privacy protection must be conditional for the vehicles which are able to be tracked and revoked if need. In addition, high mobility of vehicles and frequently changed topology are the other important characteristic of the vehicular network. Therefore, it is very challenging to design an efficient privacy preserving anonymous authentication scheme for VANETs.

In the past few years, many secure VANET schemes have been proposed, but there are still some unsolved problems. In [7], it is stated that distributing and searching of a huge certificate revocation list (CRL) is inevitable. The overhead of authentication will increase linearly with the increase of CRLs. The higher overhead of identifying and revoking malicious vehicles makes group signature and identity-based signature (GSIS) [8] and hybrid method [9] unsuitable for real-time VANETs. Efficiency conditional privacy preservation (ECPP) [10], proposed by Lu et al., is a relatively practical scheme which deals with the growing revocation list while achieving conditional traceability by the authorities, but it also suffers drawbacks: (1) It takes more latency time for sending and verifying the certificate and signature which are not efficient for the high speed movement of vehicles; (2) It needs large space to storage every vehicle's temporary information to reveal the malicious vehicles when the tracking phases; (3) Vehicle will interact with infrastructure unit several times during short-time anonymous key generation.

To resolve these problems, we propose a more efficient conditional private preservation scheme based on ECPP. The main contributions of this paper include the following: (1) Reducing the storage space. When dispute occurs, the centralized Trusted Authority can decrypt the real identity of the rogue vehicles just by certification. So it does not need to storage temporary information, and that will save considerable storage space. (2) Lowering down half of the interaction steps during anonymous key generation phase. When vehicles move along the road, the speed is usually high, and it needs fast interaction. Fewer interaction steps help to increase the interaction speed (3) Decreasing the computational cost and communication overhead for sending and verifying the certificate and signature. The presented performance studies and comparisons with ECPP and other related schemes demonstrate that this scheme is effective and efficient.

The remainder of the paper is organized as follows. In Section 2, the related work will be surveyed. In Section 3, system model, desired requirements in VANETs will be described. We will also review the bilinear pairing techniques in Section 4. The improved MECPP will be presented in Section 5. Section 6 will give security analysis about this protocol, followed by performance analysis in Section 7. Finally, we conclude the paper in Section 8. All the notations in the paper are defined in Table 1.

**Table 1.** Notations which are used in the paper.

| Notation | Descriptions |
|---|---|
| $OBU$ | The smart vehicle equipped with on-board unit |
| $RSU$ | Road side unit |
| $TA$ | The centralized Trusted Authority |
| $L_j$ | The location of RSU |
| $ID_i$ | The real identity of vehicle $i$ |
| $PID_i$ | The pseudo-id of vehicle $i$ |
| $Cert_i$ | The short-time certificate of vehicle $i$ |
| $X||Y$ | Concatenate operation |

## 2. Related Work

There have been many research works about anonymity authentication of VANETs in the last past years. In all existing works, pseudonym-based authentication schemes are the major approaches [6,7,11,12]. These pseudonym-based approaches can perfectly achieve privacy preservation, but the main limitation is that the TA is needed to keep the pseudonyms of each vehicle in storage space and required to change them in a frequent manner. To overcome this limitation, Ye et al. [13] proposed a conditional privacy preserving authentication approach using anonymous certificates. However, it is not efficient because of frequent interactions with infrastructures. Lu et al. [10] developed an efficient conditional privacy preservation scheme. This scheme can achieve perfect conditional privacy with bilinear pairing technology, but it still faces some limitation that takes more time with infrastructures and needs large storage space to save temporary messages.

Gamage et al. [14] gave a privacy protection scheme for VANET based on an ID-based ring signature without tracking function, hence this scheme does not achieve conditional privacy. Then, Zhang et al. [5] proposed another ID-based scheme named CPPA, but it is vulnerable to replay and non-repudiation attacks. To overcome the drawback of ID-based schemes, Shim improved a new efficient IBS scheme providing resistance against impersonation attack. However, it suffers to the modification attack which was demonstrated by Liu et al. [15]. In 2015, Bayat et al. [16] introduced an anonymous authentication scheme. High computational overhead is the main limitation of Bayat's scheme. In 2018, Yang et al. [4] proposed a message recovery authentication scheme based on certificateless signatures. This scheme achieves low communication costs and computation overhead without the bilinear pairing operations. However, this solution is one-way communication only from the vehicle to the RSU, and the vehicle cannot receive traffic information transmitted from the center.

Recently, Baker et al. [17] proposed an energy efficient routing protocol for VANET, called GreeAODV, which focus on selecting the most efficient routing path between source and destination. It estimate the total power consumption and locate the lowest energy consumption route. In order to guarantee user satisfaction, a Trusted Third Party (TTP) cloud entity and Quality of Experience (QoE) game model has been proposed [18]. TTPs, which provide an abstraction layer between the vehicular service users and providers, are well-known profitable commercial organizations that provide and sell service to users. TTPs thus will simplify the process of resource discovery and selection in a smart city. Based on a cluster-based Trusted Third Party (TTP) model, Ridhawi et al. [19] propose a continuous availability scheme for diversified cloud services targeting vehicle cloud users. In this scheme, timely and successful receipt of information were used to reduce highway accidents by providing early warning messages to nearby and distant vehicles, thereby increasing response time to emergencies.

Other studies accessed the anomaly detection schemes which can be used for data analysis, monitoring the normal behavior of road-side infrastructures and protecting the vehicle from potential attack [20]. Lazarevic et al. [21] presented several anomaly detection schemes to identify possible network intrusions. Now black-hole attacks are one of the security threats that appear in the network. In [22,23], Otoum et al. gave a black-hole detection scheme which help in monitoring the different aspect of systems such as pressure and temperature. Very recently, Otoum et al. [24] introduced an adaptively supervised and intrusion-aware data aggregation for wireless sensor clusters in critical infrastructures. This proposed scheme used an adaptive strategy to dynamically detect known and unknown intrusions, thus solving the intrusion problem.

## 3. System Model and System Security

In this section, we demonstrate the system model and the requirements of system security.

### 3.1. System Model

In this subsection, the system model of a single geographic region for the scheme is illustrated in Figure 1.

**System roles:** VANETs generally consist of vehicles equipped with wireless communication devices, which are called On Board Unit (**OBU**), infrastructure units such as Road Side Units (**RSU**s) which are located on the roadside or at a street intersection providing wireless interfaces to vehicles within their radio coverage, and a centralized Trusted Authority (**TA**) which is responsible for the RSU and OBU Registration, and what is more, recovering the vehicle's identity if it is necessary.

**Channels:** To secure the vehicular communications which are mainly used for civilian applications, we have been following assumptions about the channels:

- OBU communicates with RSU or other OBU through wireless links, which is unsecured.
- RSU is assumed to connect with the TA by wired links or any other creditable links with high bandwidth, low bit error rates and low delay.

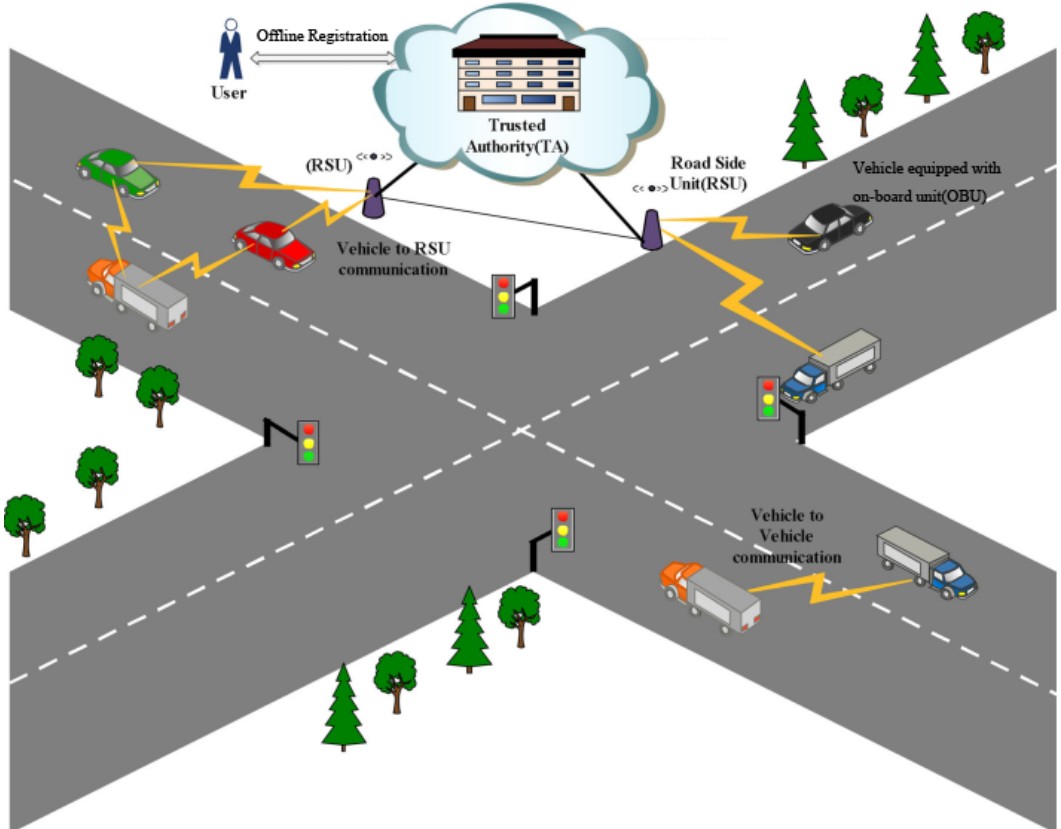

**Figure 1.** The VANET System Model. Three roles included: vehicle equipped with on-board unit (OBU), road side unit (RSU) and the trusted authority (TA).

### 3.2. System Security

In this subsection, we present the system assumption and the desired requirements for the proposed protocol.

#### 3.2.1. Secure VANETs Assumption

- All OBUs and RSUs are registered with the TA. The TA is not feasible to be compromised in the system and can be fully trusted by all parties.
- RSUs are usually deployed in open unattended environments, which can be compromised by attackers or collude with each other. However, we assume that RSUs are monitored so that their compromise can be detected in a short time. As a result, at a given time slot, very few RSUs are compromised.
- OBUs have limited computing power and storage space while TAs have greater computational power and enough hardware.

#### 3.2.2. Desired Requirements

- **Anonymous Vehicle Authentication.** The purpose of anonymous vehicle authentication is to verify a vehicle's authentic and legitimate while without revealing the real ID of the vehicle.
- **Short-term Linkability.** In some cases, like broadcasting road condition, applications require that a recipient can link two messages sent out by the same OBU in the short-term.
- **Long-term Unlinkability.** In the long-term, messages from the same vehicle should not be able to be linked by attackers or eavesdroppers.
- **Traceability and Revocation.** There must be a TA in VANETs which can trace the OBU that abuses the VANET. In addition, once the compromised OBU has been revealed, TA must revoke it immediately to prevent any further damage.

- **Non-repudiation.** Both OBUs and RSUs should not deny their behaviors and must be responsible for the decision.
- **Efficiency.** On the one hand, OBUs have resource-limited computing power to make VANETs economically viable. On the other hand, OBUs may move with the high speed. Suppose the application incorporates emergency information to be transferring to another vehicle, which has more probabilities to meet accident. This needs a quick response from the network to pass the information. A delay less than a second may cause severe damage and result in meaningless message. Therefore, the computation overhead and communication overhead at each vehicle must be as small as possible.

## 4. Preliminaries

Before presenting the scheme, we first review the pairing technique, state the definitions of the *q-SDH* assumption and weak chosen message attack.

### *4.1. Bilinear Pairing*

Let $G_1$, $G_2$ be the finite additive groups and $G_T$ be the finite multiplicative group with same order $p$ where $|G_1| = |G_2| = |G_T| = p$, the bilinear pairing $e : G_1 \times G_2 \rightarrow G_T$ satisfies the following properties [25]:

- **Bilinearity:** The mapping $e : G_1 \times G_2 \rightarrow G_T$ is said to be bilinear if the following relation holds: $e(h_1^a, h_2^b) = e(h_1, h_2)^{ab}, \forall h_1 \in G_1, \forall h_2 \in G_2$ and $\forall a, b \in Z_p$.
- **Non-degeneracy:** There exists $h_1 \in G_1, h_2 \in G_2$ such that $e(h_1, h_2)$ is not the identity of $G_T$.
- **Isomorphism:** $\psi$ is an isomorphism from $G_2$ to $G_1$, with $\psi(h_2) = h_1$
- **Computability:** The bilinear map $e : G_1 \times G_2 \rightarrow G_T$ can be computed efficiently.

### *4.2. The Strong Diffie–Hellman Assumption*

In this subsection, we state the strong Diffie–Hellman hardness assumption on which this scheme is based. Let $g_1$ be a generator of cyclic groups $G_1$ and $g_2$ be a generator of cyclic groups $G_2$. $G_1$ and $G_2$ have the same prime order $p$.

**q-Strong Diffie–Hellman Problem (q-SDH).** Given a $(q + 2)$-tuple $(g_1, g_2, g_2^x, g_2^{x^2}, ..., g_2^{x^q})$ as input, output a pair $(c, g_1^{\frac{1}{x+c}})$ where $c \in Z_p^*$. An algorithm $A$ is said to have an advantage $\varepsilon$ in solving *q-SDH* problem if

$$Pr[A(g_1, g_2, g_2^x, ..., g_2^{x^q}) = (c, g_1^{\frac{1}{x+c}})] \geq \varepsilon \tag{1}$$

where the probability is over the random choice of $x$ in $Z_p^*$ and the random bits consumed by $A$.

**Definition 1.** *We say that the* $(q, t, \varepsilon)$-*SDH assumption holds in* $(G_1, G_2)$ *if no t-time algorithm has an advantage at least* $\varepsilon$ *in solving the q-SDH problem in* $(G_1, G_2)$.

### *4.3. Weak Chosen Message Attacks*

In this paper, we will prove the scheme existential unforgeability under a weak chosen message attack [26], which need the adversary to submit all messages in advance and then are provided the public key and signatures. This notion is defined using the following game between a challenger and adversary $A$:

**Query:** A list of $q_s$ messages $M_1, ..., M_{q_s} \in \{0, 1\}^*$ were sent to challenger by the adversary $A$.

**Response:** The challenger runs algorithm *KeyGen* to generate a public key *PK* and private key *SK* and then give $A$ the public key *PK* and signatures $\sigma_i = Sign(SK, M_i)$ for $i = 1, ..., q_s$.

**Output:** Algorithm $A$ wins the game if a pair $(M, \sigma)$ is output, where:

1.  $M$ is not in $(M_1, ..., M_{q_s})$, and

2. $Verify(PK, M, \sigma) = true$

**Definition 2.** *A forger $A(t, q_s, \varepsilon)$-weakly breaks a signature scheme if A runs in time at most t, A makes at most $q_s$ signature queries, and has an advantage at least $\varepsilon$. A signature scheme is $(t, q_s, \varepsilon)$-existentially unforgeable under a weak chosen message attack if no forger $(t, q_s, \varepsilon)$-weakly breaks it.*

## 5. The Improved More Efficient Protocol

The MECPP protocol includes four parts: system initialization, temporary anonymous key generation, safe message sending, and fast tracking algorithm.

### *5.1. System Initialization*

First of all, The TA generates the system parameters $(p, G_1, G_2, G_T, g_1, g_2, e)$ for each *RSU* and vehicle using the security parameter $k$. Then, it chooses a random number $u \in Z_p^*$ as its master key and computes $U = g_2^u \in G_2$ as its public key. In addition, it selects two secure hash functions: $f$ and $h$, where $f, h : 0, 1^* \rightarrow Z_p^*$, and a secure symmetric encryption algorithm $Enc_k()$. Finally, TA publishes all public parameters $(p, G_1, G_2, G_T, g_1, g_2, e, U, f, Enc_k())$.

#### 5.1.1. OBU Registration Protocol

When an OBU register to the system with its identity $ID_i$, TA does the following:

1. Check the validity of the identity $ID_i$. If not valid, terminate the protocol;
2. Choose a fixed-length random number $rnd \in Z_p^*$, compute the pseudo-id $PID_i = Enc_u(rnd||ID_i||h(rnd||ID_i))$;
3. Set $S_i = g_1^{\frac{1}{h(PID_i)+u}} \in G_1$.
4. Return to OBU the private key $sk_i = (PID_i, S_i)$.

#### 5.1.2. RSU Registration Protocol

When an RSU applies for registering, TA does:

1. Get a location information $L_i \in Z_p^*$ such that $h(L_i) + u \not\equiv 0 \bmod p$, set $A_i = g_1^{\frac{1}{h(L_i)+u}} \in G_1$;
2. Return to RSU the location-awareness key $A_i$, where the location-awareness key means it working at location $L_i$;

Subsequently, RSU itself picks a random number $x_i \in Z_p^*$ as the secret key which is used to encrypt OBU's pseudo-id.

### *5.2. Temporary Anonymous Key Generation*

In this part, we will describe how to generate the OBU temporary anonymous key.

Based on ECPP, we propose an improved protocol. First of all, the temporary anonymous information of OBU does not have to be stored by RSU. After mutual authentication, a random pseudo-id of OBU is generated by RSU, which is contained in the temporary certificate. When a dispute occurs, the real identity of the malicious vehicles could be recovered from temporary certification by RSU and TA together. The temporary anonymous key will be changed frequently. Therefore, that will help to save large storage spaces. Secondly, the interaction rounds are decreased to three times on the premise of mutual authentication in the scheme, while six times in ECPP. Because only valid RSU at location $L_j$ can decrypt the cipher text to get the pseudo-id $PID_i$, there is no risk in disclosing its pseudo-id $PID_i$ to an attacker. It is more practical in the real world with fewer interactions. Finally, computation overhead is reduced because of less pairing operation and less point multiplication.

In Figure 2, we describe the flowchart of temporary anonymous key generation. When the vehicle enters the area of the RSU, it firstly generates its temporary short-time anonymous private

key. Then, the vehicle makes a signature as its proof and sends the request information to RSU. After receiving the request, the RSU checks the validity of signature at first. If it is valid, the OBU is authenticated. The RSU issues the temporary certificate and sends back to the vehicle. When receiving the certificate, the vehicle checks its validity. If valid, the RSU is also authenticated and the certificate is accepted. Next, we will give the detailed process of OBU temporary anonymous key generation in Table 2.

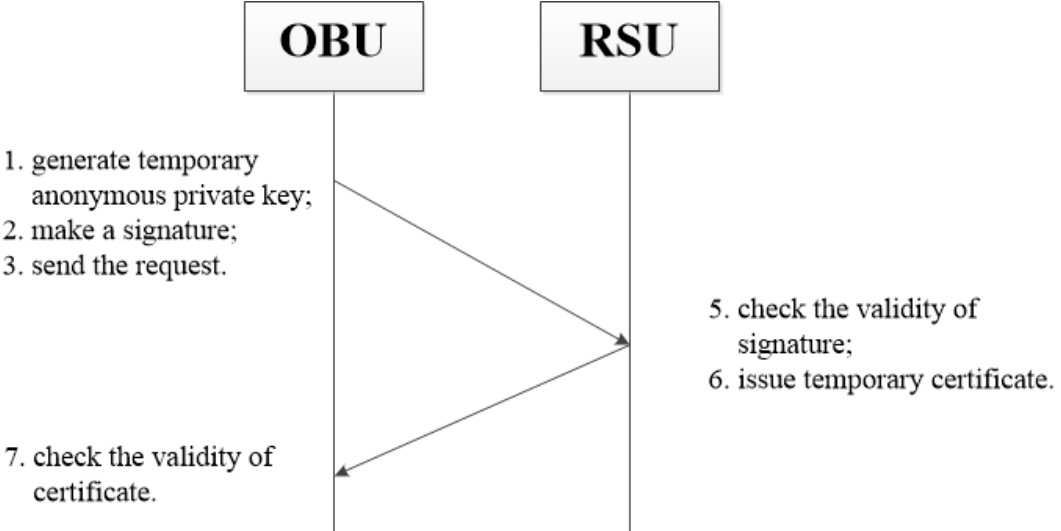

**Figure 2.** The Flowchart of Temporary Anonymous Key Generation. This flowchart describes how the OBU's temporary anonymous key is generated.

**Table 2.** OBU temporary anonymous key generation.

| OBU($ID_i, PID_i$) | RSU($ID_j$) at Location $L_j$ |
|---|---|
| $R_1 = (g_2^{h(L_j)} \cdot U)^{(r_1)}$, <br> $R_2 = e(g_1, g_2)^{r_1}$, <br> $Y = g_1^x$, <br> $Sig_{OBU} = S_i^{(r_1 + f(R_2\|\|T_i\|\|Y))}$, <br> $C = Enc_{R_2}(Y, T_i, Sig_{OBU}, PID_i)$. | |
| $\xrightarrow{\quad (R_1, C) \quad}$ | |
| | $R_2' = e(A_j, R_1)$, <br> decrypt $C$ as $Dec_{R_2'}(C)$, judge $T_i$ and $PID_i$, <br> check <br> $R_2' \cdot e(g_1, g_2)^{f(R_2'\|\|T_i\|\|Y)} \overset{?}{=} e(Sig_{OBU}, g_2^{h(PID_i)} \cdot U)$, <br> issue the certificate <br> $Cert_i = (L_j, T_i, Y, PID_i', Sig_{RSU})$, <br> where <br> $PID_i' = Enc_{x_j}(T_i, PID_i)$ and <br> $Sig_{RSU} = A_j^{f(R_2'\|\|T_i\|\|Y\|\|PID_i')}$. |
| $\xleftarrow{\quad (Cert_i) \quad}$ | |
| Judge $T_i$ and check <br> $e(g_2^{h(L_j)} \cdot U, Sig_{RSU}) \overset{?}{=} e(g_1, g_2)^{f(R_2\|\|T_i\|\|Y\|\|PID_i')}$. | |

- Step 1. When an OBU goes into the location $L_j$, it firstly computes $R_1 = (g_2^{h(L_j)} \cdot U)^{(r_1)} \in G_2$ and $R_2 = e(g_1, g_2)^{r_1}$ where $r_1 \in Z_p^*$ is a random number. Then, the OBU chooses another random number $x \in Z_p^*$ as its temporary short-time anonymous private key, computes the corresponding temporary public key $Y = g_1^x \in G_1$. At last, the OBU uses its private key $S_i$

to make a signature $Sig_{OBU} = S_i^{(r_1+f(R_2||T_i||Y))}$ where $T_i$ is the current time-stamp, encrypts the signature as $C = Enc_{R_2}(Y, T_i, Sig_{OBU}, PID_i)$, and sends request information $(R_1, C)$ to the $RSU(ID_j)$.

- Step 2. After receiving the request, $RSU(ID_j)$ computes $R_2' = e(A_j, R_1)$, and decrypts the cipher text $C$ with $R_2'$. Then, $RSU(ID_j)$ will check the validity of $T_i$ and $PID_i$. Either of them are invalid, the protocol aborts. Otherwise, $RSU(ID_j)$ checks the equation $R_2' \cdot e(g_1, g_2)^{f(R_2'||T_i||Y)} \stackrel{?}{=} e(Sig_{OBU}, g_2^{h(PID_i)} \cdot U)$. If it holds, i.e., the OBU is authenticated, then $RSU(ID_j)$ issues the certificate $Cert_i = (L_j, T_i, Y, PID_i', Sig_{RSU})$, where $PID_i' = Enc_{x_j}(T_i, PID_i)$ and $Sig_{RSU} = f(R_2'||T_i||Y||PID_i')A_j$, the lifecycle of certification is based on time-stamp $T_i$; otherwise, the OBU fails the authentication since

$$
\begin{aligned}
e(Sig_{OBU}, g_2^{h(PID_i)} \cdot U) &= e(S_i^{(r_1+f(R_2||T_i||Y))}, g_2^{h(PID_i)} \cdot g_2^u) \\
&= e(g_1^{\frac{(r_1+f(R_2||T_i||Y))}{h(PID_i)+u}}, g_2^{(h(PID_i)+u)}) \\
&= R_2' \cdot e(g_1, g_2)^{f(R_2'||T_i||Y)}
\end{aligned}
\tag{2}
$$

- Step 3. To verify $RSU(ID_j)$ and the validity of certificate $Cert_i$, the OBU checks $e(g_2^{h(L_j)} \cdot U, Sig_{RSU}) \stackrel{?}{=} e(g_1, g_2)^{f(R_2||T_i||Y||PID_i')}$. If it holds, $Cert_i$ is valid and the RSU is also authenticated, because the adversary has no ability to recover the secret key $R_2$; Otherwise, the protocol aborts and the RSU cannot pass the authentication since

$$
\begin{aligned}
e(g_2^{h(L_j)} \cdot U, Sig_{RSU}) &= e(g_2^{h(L_j)} \cdot g_2^u, A_j^{f(R_2'||T_i||Y||PID_i')}) \\
&= e(g_2^{(h(L_j)+u)}, g_1^{\frac{f(R_2'||T_i||Y||PID_i')}{h(L_j)+u}}) \\
&= e(g_1, g_2)^{f(R_2'||T_i||Y||PID_i')} \\
&= e(g_1, g_2)^{f(R_2||T_i||Y||PID_i')}
\end{aligned}
\tag{3}
$$

### 5.3. Safe Message Sending

1. Signing: When a vehicle *i* wants to send message *M* to other surrounding vehicles, it signs on message *M* with the short-time anonymous public-key certificate $Cert_i$ and the private key *x* before sending it out.

   - *Step 1.* Compute $R = g_1^r \in G_1$ where $r \in Z_p^*$ is a random number, and sign the message $s_r \equiv r + x \cdot h(M, R) \pmod{p}$.
   - *Step 2.* Set signature $Sig_M = (R, s_r, Cert_i)$.

2. Verification: Once receiving the message, the receiver is firstly checking the validity of $T_i$ and $Cert_i$ like Step 3 in Section 5.2. If invalid, the verification process aborts. Otherwise, the receiver verifies the signature $Sig_M$ by checking the equation $g_1^{s_r} = R \cdot Y^{h(M,R)}$. If it holds, the message is true and can be accepted, otherwise neglected, since

$$
\begin{aligned}
R \cdot Y^{h(M,R)} &= g_1^r \cdot g_1^{x \cdot h(M,R)} \\
&= g_1^{r+x \cdot h(M,R)} \\
&= g_1^{s_r}
\end{aligned}
\tag{4}
$$

### 5.4. Fast Tracking

Tracing operation is an essential issue for anonymous communication system. If a malicious vehicle makes a violation, the real identity of the signature should be revoked and transferred to the judiciary for punishment. When the TA receives the report:

- *step 1.* The TA sends the tracing demand $(M, Sig_M)$ to the specified RSU according to the location information $L_j$ in $Cert_i$.
- *step 2.* The RSU returns the pseudo-id $PID_i$ to TA by decrypting $PID_i = Dec_{x_j}(PID_i')$ with security key $x_j$.
- *step 3.* The TA recovers the real identity $ID_i$ by decrypting $rnd||ID_i||h(rnd||ID_i) = Dec_u(PID_i)$ with master key $u$ and then calculate $h'(rnd||ID_i)$. If $h'(rnd||ID_i) = h(rnd||ID_i)$, the $ID_i$ and $PID_i$ are valid and then broadcasts the pseudo-id $PID_i$ to all RSUs. Then, the malicious vehicle cannot get temporary short-time anonymous key from the RSUs any more.

## 6. Security Analysis

In this section, we analyse the security of the proposed scheme. First, we will describe the provable security. Next we analyse more security requirements proposed in Section 3.2.

### 6.1. Provable Security

**1. Private Key Security.** The TA use master key to allocate initial private keys to OBUs or RSUs during the registration stage. The security of private key is based on the *q-SDH* [27] hardness assumption. Even through several OBUs and RSUs are compromised, deducing the private keys of other OBUS and RSUs from the compromised private key is still computationally not feasible. It is still computationally not feasible to deduce other OBUs and RSUs' private keys from the compromised private keys.

**Lemma 1.** *If the q-SDH assumption holds in $(G_1, G_2)$, then this scheme is secure against existential forgery under a chosen message attack.*

**Proof of Lemma 1.** Assume $A$ is a forger that $(t, q_S, \varepsilon)$-breaks the scheme and $B$ is an attacker which solves the *q-SDH* problem in time $t'$ with advantage $\varepsilon$ by interacting with $A$. $(g_1, g_2, A_1, ..., A_q)$ is an instance of the *q-SDH* problem, where $A_i = g_2^{(x^i)} \in G_2$ for $i = 1, ..., q$ and for some unknown $x \in Z_p^*$. For convenience, we set $A_0 = g_2$. Algorithm $B$'s goal is to produce a pair $(c, g_1^{\frac{1}{x+c}})$ for some $c \in Z_p^*$. It does so as follows:

**Query:** Algorithm $A$ chooses a list of random pseudo-id $PID_1, PID_2, ..., PID_{q_s} \in Z_p^*$, and requests for private key of $PID_i$, where $q_s < q$. We may assume that $q_s = q - 1$.

**Response:** $B$ must response with $TA$'s public key and $PID_i$'s private keys. Let $f(y)$ be the polynomial $f(y) = \prod_{i=1}^{q-1}(y + h(PID_i))$. Expand $f(y)$ and write $f(y) = \sum_{i=0}^{q-1} \alpha_i y^i$ where $\alpha_0, ..., \alpha_{q-1} \in Z_p$. Compute:

$$P_2' \leftarrow \prod_{i=0}^{q-1}(A_i)^{\alpha_i} = g_2^{f(x)} \quad and \quad K_{TA} \leftarrow \prod_{i=1}^{q}(A_i)^{\alpha_{i-1}} = g_2^{xf(x)} = (g_2')^x \tag{5}$$

Also, let $P_1' = \psi(P_2')$. The public key given to $A$ is $(P_1', P_2', K_{TA})$. Next, algorithm $B$ will generate private keys $k_i$ for each $PID_i$ where $i = 1, 2, ..., q - 1$. To do so, let $f_i(y)$ be the polynomial $f_i(y) = f(y)/(y + h(PID_i)) = \prod_{j=1, j \neq i}^{q-1}(y + h(PID_j))$. We expand and write $f_i(y) = \sum_{j=0}^{q-2} \beta_j y^j$. Compute

$$S_i \leftarrow \prod_{j=0}^{q-2} A_j^{\beta_j} = g_2^{f_i(x)} = (g_2')^{\frac{1}{x+h(PID_i)}} \in G_2 \tag{6}$$

Observe that $k_i = \psi(S_i) \in G_1$ is a valid private key of $PID_i$ under the public key $(P_1', P_2', K_{TA})$. Algorithm $B$ gives the $q - 1$ private keys $k_1, ..., k_{q-1}$ to $A$.

**Output:** Algorithm $A$ returns a forgery $(PID_*, k_*)$ such that $k_* \in G_1$ is a valid private key for $PID_*$ and $PID_* \notin PID_1, ..., PID_{q-1}$. In other words, $e(k_*, K_{TA} \cdot (g_2')^{h(PID_*)}) = e(g_1', g_2')$. Since $K_{TA} = (g_2')^x$, we have that $e(k_*, (g_2')^{(x+h(PID_*))}) = e(g_1', g_2')$ and therefore

$$k_* = (g_1')^{\frac{1}{x+h(PID_*)}} = g_1^{\frac{f(x)}{x+h(PID_*)}} \tag{7}$$

Using long division, we expand the polynomial $f$ as $f(y) = \gamma(y)(y + h(PID_*)) + \gamma_{-1}$ for some polynimal $\gamma(y) = \sum_{i=0}^{q-2} \gamma_i y^i$ and some $\gamma_{-1} \in Z_p$. Then, computing as

$$f(y)/(y + h(PID_*)) = \frac{\gamma_{-1}}{y + h(PID_*)} + \sum_{i=0}^{q-2} \gamma_i y^i \tag{8}$$

Note that $\gamma_{-1} \neq 0$, since $f(y) = \prod_{i=1}^{q-1}(y + h(PID_i))$ and $PID_* \notin PID_1, ..., PID_{q-1}$, as thus $(y + h(PID_*))$ does not divide $f(y)$. Then, algorithm $B$ computes

$$\omega \leftarrow (k_* \cdot \prod_{i=0}^{q-2} \psi(A_i)^{-\gamma_i})^{1/\gamma_{-1}} = g_1^{\frac{1}{x+h(PID_*)}} \tag{9}$$

and returns $(h(PID_*), \omega)$ as the solution to the $q - SDH$ instance. $\square$

**2. Signature Security.** The security of OBU's signature $Sig_M$ is based on the discrete logarithm assumption. it is not feasible to output a forgery in polynomial time, which makes the scheme resistive to the impersonation attack and the bogus message spoofing attack.

**Lemma 2.** *If the discrete logarithm assumption holds, then the signature is secure against existential forgery under an adaptively chosen message attack.*

**Proof of Lemma 2.** We suppose that $A$ which is an adversary taking message $M$ and public key $Y$ as input has a non-negligible probability to output an existential forgery in polynomial time. Then, $A$ can get two forgeries for the same message according to the forking lemma [28]. Let $Sig_M = (R, s_1)$ and $Sig_M' = (R, s_2)$ are the two signature forgeries, respectively, where $R = g_1^r, s_1 = r + x \cdot h(M, R) \bmod p$ and $s_2 = r + x \cdot h'(M, R) \bmod p$. Then, we have the following equation.

$$s_1 - s_2 = x(h(M, R) - h'(M, R)) \quad mod \quad p \tag{10}$$

Hence

$$x = (s_1 - s_2)(h(M, R) - h'(M, R))^{-1} \quad mod \quad p \tag{11}$$

As can be seen from the above, $x$ can be computed successfully, but it contradicts the discrete logarithm assumption. Therefore, $Sig_M$ is difficult to be forged. $\square$

*6.2. Further Security Analysis of The Proposed Scheme*

**1. Mutual Authentication**. The scheme realizes mutual authentication between the RSU and the OBU by the request-response protocol.

- *The RSU can quickly authenticate the OBU.* In Step 2 of Section 5.2, if the verification equation $R_2' \cdot e(g_1, g_2)^{f(R_2'||T_i||Y)} = e(Sig_{OBU}, g_2^{h(PID_i)} \cdot U)$ holds, the OBU can be authenticated with pseudo-id $PID_i$. Since the private key is secure according to **Lemma 1**, therefore, $Sig_{OBU}$ is unforgeable, and no adversary can launch an impersonation's attack on the RSU.
- *The OBU can also efficiently authenticate the RSU at location $L_j$.* In Step 3 of Section 5.2, if the equation $e(h(L_j)P_2 + U, Sig_{RSU}) = e(g_1, g_2)^{f(R_2||T_i||Y||PID_i')}$ holds, the RSU is authenticated. Because the adversary is not feasible to recover the correct $R_2$ without knowing the RSU's private key $A_j = g_1^{\frac{1}{h(L_j)+u}}$.

**2. Anonymous Vehicle Authentication**. The OBU's identity can be kept perfectly anonymous in this protocol, since the real ID of OBU is not known to the RSU and other vehicles except the TA.

- When the OBU requests for a short-time anonymous key, it sends to RSU the pseudo-id $PID_i = Enc_u(rnd||ID_i)$ which is a random identity mark, and RSU does not know which it is.
- When OBUs communicate each other, OBU uses a random pseudo-id $PID_i' = Enc_{x_j}(T_i, PID_i)$ to denote the identity, it is different with time going by and it has no means to other OBUs.

**3. Short-term Linkability**. Since the anonymous key is valid for a short time interval, any message signed by that key can be linked.

**4. Long-term Unlinkability**. In order to protect the privacy of the driver, we require that the information sent by the same vehicle be unlinkable in the long-term. We calculate the probability to quantify the risk that the victim OBU is tracked by some compromised RSUs. Here, we give some assumptions:

- The RSUs may be compromised because of the insecure environment, but will be quickly rescued in the next period. We assume that the number of RSUs is $N_{rsu}$, and that $p_c$ RSUs can be compromised. Then, the number of compromised RSUs is $N_c = N_{rsu} * p_c$.
- We assume that the number of anonymous keys that an OBU requests at some period is $N_k$.

Let $Pr\{i\}$ represent the probability that exactly $i$ ($i \geq 2$) among $N_k$ anonymous keys are requested from different compromised RSUs, we have $Pr\{i\} = \frac{\binom{N_{rsu}-N_c}{N_k-i}\binom{N_c}{i}}{\binom{N_{rsu}}{N_k}}$. Then, the probability is

$$Pr\{i \geq 2\} = 1 - Pr\{i = 0\} - Pr\{i = 1\}$$
$$= 1 - \frac{\binom{N_{rsu}-N_c}{N_k}\binom{N_c}{0} + \binom{N_{rsu}-N_c}{N_k-1}\binom{N_c}{1}}{\binom{N_{rsu}}{N_k}} \tag{12}$$

From Figure 3 below, it can be seen that the tracking probability increases very slowly with the increase of the number of anonymous keys and the number of compromised RSUs. So it is long-term unlinkability.

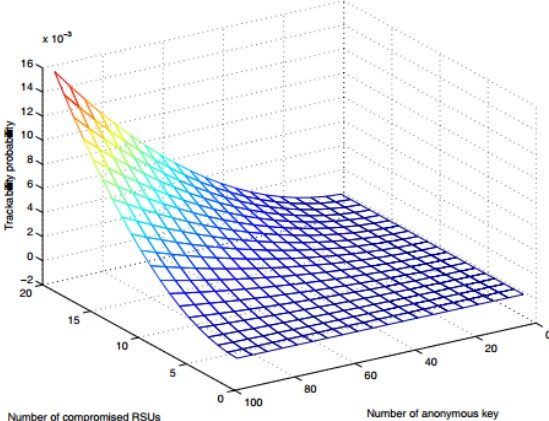

**Figure 3.** Tracking Probability of the system. Note that it increases very slowly with the increase of the number of anonymous keys and the number of compromised RSUs

**6. Traceability**. Even if the message does not contain identifying information about vehicles, by using the Fast Tracking algorithm described in Section 5.4, the TA can recover the real identity of the malicious vehicle if required.

**7. Non-repudiation**. It is obvious that signature $Sig_{OBU}$ of OBU can provide the non-repudiation proof on the OBU's temporary anonymous key requesting, while signature $Sig_{RSU}$ of RSU provide the non-repudiation proof on cert issue.

## 7. Performance Analysis

In this section, we compare the performance of the proposed protocol with other related schemes in terms of computational cost, communication overheads and storage overhead. To measure the effectiveness of the proposed scheme, we present the comparison results in different tables.

### 7.1. Computational Cost Analysis

We evaluate the computational cost for anonymous certificate and signature verification process with the prior related schemes such as ECPP [10], Boneh Lynn Shacham (BLS) scheme [29], group signature based (GSB) scheme [8], certificateless aggregate signatures (CAS) [30] and key-insulated pseudonym self delegation (KPSD) scheme [31]. To facilitate analysis, the using notations are given in Table 3, and the result was obtained on a 2.6-GHz computer with 8-GB installed memory, running Cygwin 1.7.35-15 [32] with the gcc version 4.9.2 for the implementations.

**Table 3.** The notation and processing time for computational cost analysis.

|  | Descriptions | Execution Time |
|---|---|---|
| $T_{pmul}$ | The time for one point multiplication | 0.6 ms |
| $T_{pair}$ | The time for one pairing operation | 1.6 ms |
| $T_{hash}$ | The time for one hash function | 2.7 ms |
| $T_{exp}$ | The time for one exponentiation operation | 0.6 ms |

The results of computation cost comparisons with ECPP, BLS, GSB, CAS, KPSD schemes are summarized in Table 4. From Table 4, it can be clearly observed that the proposed MECCP scheme takes less computational cost than others, because it takes only $2T_{pair}$, $T_{exp}$ and $T_{hash}$ for verifying a certificate and signature. Especially comparing with ECPP, the required execution time has been decreased by about 54%.

**Table 4.** Computational cost between the proposed scheme and other related schemes.

| Scheme | Certificate and Signature | Execution Time |
|---|---|---|
| ECPP | $3T_{pair} + 11T_{pmul} + T_{hash}$ | 14.1 ms |
| BLS | $4T_{pair} + 2T_{hash}$ | 11.8 ms |
| GSB | $3T_{pair} + 9T_{exp} + T_{hash}$ | 12.9 ms |
| CAS | $5T_{pair} + 2T_{hash}$ | 13.4 ms |
| KPSD | $4T_{pair} + 10T_{exp} + T_{hash}$ | 15.1 ms |
| MECPP(Proposed) | $2T_{pair} + T_{exp} + T_{hash}$ | 6.5 ms |

### 7.2. Communication Overheads Analysis

In this section, we discuss the communication cost of the proposed scheme with ECPP. To facilitate comparisons, we assume that the bit length of random number and time stamp are 4 bytes, the bit length of ID and PID are 20 bytes, the bit length of the elements in $G_1$ and $G_2$ are 20 bytes. Furthermore, assume that the bit length of the signature messages are all the same. In Table 5, we summarize the communication overhead of sending the signature message between the proposed scheme and ECPP.

**Table 5.** Communication overhead between the proposed scheme and efficiency conditional privacy preservation (ECPP).

| Scheme | Sending the Signature Message | Size of Signature Message |
|---|---|---|
| *ECPP* | $(\sigma_M, Y, T_i, Cert_{t_i})$ | 120 bytes |
| *MECPP(Proposed)* | $(R, s_r, Cert_{t_i})$ | 108 bytes |

In the proposed scheme, the total size of the signature message $(R, s_r, Cert_{t_i})$ is $20 + 4 + 20 + 4 + 20 + 20 + 20 = 108$ bytes where $Cert_i = (L_j, T_i, Y, PID'_i, Sig_{RSU})$, while the size of the signature message $\sigma_M, Y, T_i, Cert_{t_i}$ in ECPP is $24 + 20 + 4 + 20 + 20 + 20 + 4 + 4 + 4 = 120$ bytes where $Cert_{t_i} = (T_U, T_V, c, s_a, s_x, s_\delta)$.

Besides, the interaction steps during temporary anonymous key generation phase is three times while ECPP is six times.

From the analysis of communication cost above, it can be seen that the communication overhead has been reduced by about 10% and the proposed scheme is more efficient than ECPP.

*7.3. Storage Analysis*

In this section, we analyse the storage cost between the proposed scheme and ECPP. To track the malicious vehicle, some information of vehicles should be saved by TA and RSU in ECPP. These will take a large of storage space especially the information of temporary anonymous keys, because the temporary anonymous keys of vehicles will be changed frequently to hide the real identity of vehicles. While in the proposed scheme, the real identity could be recovered by TA and RSU together from the certificate if necessary. Table 6 shows the storage cost between the scheme and ECPP.

**Table 6.** Storage cost between the proposed scheme and ECPP.

| Scheme | Stored Message about Temporary Anonymous Key | Size of Stored Message |
|---|---|---|
| *ECPP* | $PID_i, T_i, Y, R_2, \sigma_1$ | 64 bytes |
| *MECPP(Proposed)* | —- | —- |

Considering that the temporary anonymous key will be changed frequently to secure the identity, it helps to save a large of storage space for RSU. In this sense, the MECPP protocol is more practical than ECPP.

## 8. Conclusions

In this paper, we proposed a new optimized protocol called MECPP based on ECPP for secure vehicular communications. The proposed MECPP scheme achieves mutual authentication between the RSUs and the vehicles in an anonymous manner before temporary certification. Particularly, the MECPP scheme not only provides the security and privacy protection to vehicles, but also provides a fast tracking mechanism to reveal the real identity of the malicious vehicles. In addition, the performance analysis section shows that the proposed scheme outperforms the ECPP scheme in terms of computational cost, communication overheads and storage overhead.

As future work, we propose to continue to optimize the latency, such as reducing communication overhead. Furthermore, we will aim to develop a new batch authentication scheme to simultaneously authenticate the vehicles in order to avoid the computation burden in large vehicular clouds.

**Author Contributions:** T.W. conceived the ideas. T.W. performed the theoretical results and carried out the numerical simulation supervised by X.T.

**Funding:** This research received no external funding.

**Conflicts of Interest:** The authors declare no conflict of interest.

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
