# Peer review of "A More Efficient Conditional Private Preservation Scheme in Vehicular Ad Hoc Networks"

_applsci, doi:10.3390/app8122546_

Reviewer 1 Report

The authors in this paper have extended the work of Lu et al. (i.e. ECPP: Efficient Conditional Privacy Preservation Protocol for 327 Secure Vehicular Communications) to improve low storage space and enhance computational cost issues. Lu has proposed an protocol based conditional privacy preservation, ECPP for vehicles in VANETs. The work done in this paper is promising, however, cannot be accepted in its current format. Major revision required as follows,

1. Abstract needs more information on your proposal improved Lu overall performance. Mentioned some numbers or statistic to make the reader better convince in your results.

2. Lots of work related to VANET have been already disseminated in the literature. Thus, your introduction is short and need to be more comprehensive. Some of the studies that can help you:

- "Vehicle as a resource for continuous service availability in smart cities."

- "Detection of Known and Unknown Intrusive Sensor Behavior in Critical Applications."

- "On the impact of quality of experience (QoE) in a vehicular cloud with various providers."

3.  The authors have conducted NO literature review section (Very Very short!). Some very important references are still missing. The following references should be included in the revision which is very recent and with the touch of deep learning and ML related to security in critical infrastructures. while another reference to give more information about what is the development on vehicular cloud recently:

- "GreeAODV: An Energy Efficient Routing Protocol for Vehicular Ad Hoc Networks."

- "Multiagent/multiobjective interaction game system for service provisioning in vehicular cloud."

- "Hierarchical trust-based black-hole detection in WSN-based smart grid monitoring."

- "A novel self-adaptive content delivery protocol for vehicular networks"

- "Adaptively Supervised and Intrusion-Aware Data Aggregation for Wireless Sensor Clusters in Critical Infrastructures"

- "Security threats to critical infrastructure: the human factor."

- "Congestion Mitigation in Densely Crowded Environments for Augmenting QoS in Vehicular Clouds." "

- "Mitigating False Negative intruder decisions in WSN-based Smart Grid monitoring." Wireless Communications and Mobile Computing Conference "

- "A continuous diversified vehicular cloud service availability framework for smart cities."

- "Fairness-Aware Game Theoretic Approach for Service Management in Vehicular Clouds." 

"A generalized framework for quality of experience (QoE)-based provisioning in a vehicular cloud."

4. The authors should increase the technical write up of their work. The paper length seems to be short for a journal article.

5. Line 62 is odd. Just add it to the figure caption!

6. In section 5.3 Safe Message Sending. How do you compute R. Not mentioned to the process?

7. Performance Analysis section is not mature. You need to conduct a real simulation to support your mathematical computations. At least you should simulate Low storage and cost functions!

8. Since this is an extension work from another one, you are highly suggested to add a table of similarities and differences. What have you done and improved to Liu work!

 Author Response

Thank you very much for your valuable and insightful comment. Please see the attachment for response.

Thanks again and have a good day.

Tao Wang

Reviewer 2 Report

The paper faces the problem of conditional anonymous authentication, which is an important topic in future vehicular networks.

The proposed solution is an extension of ECPP presented in [8] to reduce the storage space, the number of interactions steps and the overhead computation.

Results show the improvement of the proposed solution with respect to its reference ECPP.

However, I have some comments. Going by order:

-      I suggest to not introduce citations in the abstract.

-      The context of the work should be better individuated. Authors write about “wonderful” future applications, but they do not specify when and what in the different countris is going to happen. Some works could be considered in this context. See, e.g., A Survey on the Roadmap to Mandate on Board Connectivity and Enable V2V-Based Vehicular Sensor Networks, 2018 or A Survey of the Connected Vehicle Landscape--Architectures, Enabling Technologies, Applications, and Development Areas, 2017.

 -      Some typos and spaces among words must be corrected.

 -      Section 2 is really poor. There are a lot of related works on connected vehicles and privacy issues. And a wider investigation should be provided. Just to give few examples: A Survey on Infrastructure-Based Vehicular Networks, 2017 or Security and privacy in vehicular networks, 2015 or Measuring Privacy in Vehicular Networks, 2017

 -      Some theorems or lemmas are proved. Some other no. Authors should provide the proof of all the presented theorems. For example, what about Theorem 2 demonstration?

 -      Results are presented just in terms of tracking probability and in a simple comparison with ECPP. But what about other known scheme? At least a discussion about the proposed solution performance and other solutions should be given. 

Author Response

Thank you very much for your valuable and insightful comment. Please see the attachment for response.

Thanks again and have a good day.

Tao Wang

Reviewer 3 Report

This paper proposes an improved protocol based on the concept of ECPP 

which uses minimal interaction steps, little storage space and less computation overhead to achieve more efficiency conditional privacy preservation(MECPP) scheme in VANETs.

RESEARCH ISSUES

- Add some king of sequence diagrams to make the paper more readeable. Table 1 is ok, but you can add some king of diagram to explain it better.

Specially in section 5 to get an overall view of the proposed model.

- PAGE 2 (figure 1) - Try to use a more beautiful picture. Just some samples (but make sure you have copyright permission):

https://ubisafe.org/explore/congested-clipart-traffic-car/

https://media.treehugger.com/assets/images/2017/06/hyperlane-croped.jpg.860x0_q70_crop-scale.jpg

- PAGE 2: Poor related work, (or it is also in the introduction). Join Introduction together with Related Work. 

- PAGE 4: A proof of theorems 1 and 2 ??? (or they are no theorem?)

- Please write something in section 3 to introduce subsections and/or to explain the following subsections.

- Please write something in section 4 to introduce subsections and/or to explain the following subsections.

- Please write something in section 6 to introduce subsections and/or to explain the following subsections.

- Add detail information in figure and tables caption (as an example in figure 2 you mention "Tracking Probability", it is better to mention "... Tracking probability of the system. Note that it increases very slowly with the  increase of the number of anonymous keys and the number of compromised RSUs ..." or something similar)

- I recomend you to add more up-to-date bibliography.

- Add a section

  The following abbreviations are used in this manuscript:

- I think this paper has a weak performace analysis ... try to compare to ECPP protocol. As you state in the abstract " ... which uses minimal interaction steps, little storage space and less computation overhead to achieve more efficiency conditional privacy preservation .... ". Include a table, figure, some paragraphs to justify such statement.

FORMATING ISSUES

- I think you have use the Word template, since inline equations are ... a bit (how to say) .... you know.

- Use or check the MDPI LaTeX template or Word template and include the following issues:

- in the author name Xiaohu Tang ... after the superscript 1 there is a comma ,

- Duplicate .... School of Information Science and Technology, Southwest Jiaotong University, Chengdu 610000, China

- keywords ... all in lower

- Current address: Affiliation 3 ????

- submitted to Journal Not Specified ??? (add the journal name)

- Some section / subsection titles are in Capitals, why? (see section 2, 7, 8, etc...) Review the capitals in section/subsections.

- PAGE 10: Include Funding (mandatory), if no funding please say it and Acknowledgment if needed.

- LAST PAGE: Remove Sample Availability (if needed) or complete the sentence (.....)

Author Response

Thank you very much for your valuable and insightful comment. Please see the attachment for response.

Thanks again and have a good day.

Tao Wang

Round  2

Reviewer 1 Report

I have noticed that the authors addressed my comments. I would like them to go over the following minor comments as wee: 

1. Figure 1 is big. You need to reduce it a bit since it consumes a big space. 

2. Related work section still very small. You ignored most of the suggested work in my initial review. Please re-consider the following and especially the security aspect as your work consider a critical infrastructure:

- GreeAODV: An Energy Efficient Routing Protocol for Vehicular Ad Hoc Networks." 

- "Hierarchical trust-based black-hole detection in WSN-based smart grid monitoring." 

- "Adaptively Supervised and Intrusion-Aware Data Aggregation for Wireless Sensor Clusters in Critical Infrastructures" 

- "Congestion Mitigation in Densely Crowded Environments for Augmenting QoS in Vehicular Clouds." - "Security threats to critical infrastructure: the human factor." 

- "Mitigating False Negative intruder decisions in WSN-based Smart Grid monitoring." Wireless Communications and Mobile Computing Conference "

- "A continuous diversified vehicular cloud service availability framework for smart cities." 

- "Fairness-Aware Game Theoretic Approach for Service Management in Vehicular Clouds." 

3. Table 2 is the core of your key generation, can you add a flowchart to help the reader navigate through it? 

4. The conclusion is not comprehensive, Need serious attention. Future work as well!

Author Response

Dear Reviewer

Thanks for your positive comments and valuable suggestions to help us improve the paper. Please check the attachment for detailed point-by-point  response.

Reviewer 2 Report

The authors worked on the paper, trying to follow most of the reviewers' comments and improving  some aspects of the work.

However, exception made for one, the authors did not take into account the suggestions of all the reviewers on the bibliography, which, instead remains poor.

Since a reviewer dedicate time in reviewing and suggesting related work, it would be appreciated if authors could take into account the suggestions.

Author Response

(The authors gave the same response as above.)

Reviewer 3 Report

RESEARCH ISSUES

This paper prososes an improved protocol based on the concept of ECPP protocol has been proposed to achieve more efficiency conditional privacy preservation (MECPP) scheme in VANETs.

All previous comments (round 1) has been solved.

ENGLISH ISSUES (Minor issues)

Line 2.- "...networks(VANETs)..." add a blank "...networks (VANETs)..."

and in all acronyms along the paper (line 36, etc..)

Lines 60-61 (Table 1.-). Dont use "... our paper...", replace by "...the paper"/"...this paper", use impersonal.

Lines 7, 55, 58, 59, 60, 87, 131, 143, 153, 166, 167, 197, 233, 242, 244, 268, 281, etc.... ".. our ..."

Line 100.. "... our proposed ..."  use impersonal.

LAYOUT ISSUES

Table 2.- see right margin of the table. I think that in the proof stage it will be corrected by editors.

Author Response

(The authors gave the same response as above.)
